# OpenReview forum: "HaluEval-Wild: Evaluating Hallucinations of Language Models in the Wild"
_ICLR.cc/2025/Workshop/BuildingTrust — BuildingTrust_

### Official Review · Reviewer_CRWM · 2025-02-24

**Rating:** 6
**Confidence:** 4

**Review:**

### Summary
This paper introduces HaluEval-Wild, a benchmark for evaluating hallucinations in LLMs using real-world user queries from ShareGPT. The dataset consists of 500 adversarially filtered queries, categorized into five types (Out-of-Scope, Complex Reasoning, Inappropriate Content, Beyond-Modality, and Confused Queries).
The study compares hallucination rates across multiple LLMs (GPT-4, GPT-3.5, Mixtral, Mistral, Llama-2, Vicuna, and Alpaca) and finds that knowledge-distilled models tend to hallucinate more. It also evaluates Self-Reflection (SR) and Retrieval-Augmented Generation (RAG), demonstrating that RAG reduces hallucination rates from 20% to 5% in GPT-4.
The benchmark, available on GitHub, provides a real-world evaluation framework for improving LLM reliability.

### Strongness
- Unlike traditional hallucination benchmarks, this study leverages real-world user queries from ShareGPT, making the evaluation more reflective of practical challenges faced by LLMs.
- The five-category classification of hallucination-prone queries enables a fine-grained analysis, allowing researchers to understand the weaknesses of LLMs across different types of challenges.
- The paper rigorously tests Self-Reflection (SR) and RAG, demonstrating their effectiveness in reducing hallucinations. This provides practical guidance for improving LLM reliability.

### Weakness
- The study evaluates a range of LLMs, but lacks results for state-of-the-art proprietary models such as Claude and Gemini, which could provide a more complete picture of hallucination rates.
- While RAG significantly reduces hallucination rates, its effectiveness depends on the quality and recency of retrieved information. The paper does not discuss potential biases or limitations introduced by retrieval-based methods.

---

### Official Review · Reviewer_WrTC · 2025-03-01
**The paper presents HaluEval-Wild, a benchmark for evaluating hallucinations in LLMs using real-world queries, highlighting its structured categorization and mitigation strategies while noting limitations in model choices, dataset size, and reference answer bias.**

**Rating:** 6
**Confidence:** 3

**Review:**

Summary [This paper introduces HaluEval-Wild, a benchmark designed to evaluate hallucinations in large language models (LLMs) in real-world settings. The authors curate 500 challenging user queries from ShareGPT, filtering them using Alpaca and categorizing them into five types. They generate reference answers using GPT-4 with retrieval-augmented generation (RAG) and evaluate various LLMs on their hallucination rates. The study also explores self-reflection techniques for mitigating hallucinations.]

Strengths
[Real-World Focus: Unlike traditional hallucination benchmarks, HaluEval-Wild captures user interactions in the wild, making it more relevant for practical applications.
Structured Categorization: The classification of hallucinations into five distinct types allows for a more fine-grained analysis of model failures.
Evaluation Across Multiple Models: The paper provides comparative hallucination rates for various open-source and closed-source LLMs.
Mitigation Strategies: The study tests self-reflection techniques, providing insights into how LLMs can be improved to reduce hallucinations.
Comprehensive Dataset Collection Pipeline: The methodology, including adversarial filtering with Alpaca and manual verification, ensures that the dataset contains genuinely challenging queries.]

Weaknesses
[Justification for Model Choices:
The paper uses Llama 2-7B to classify queries but Alpaca for adversarial filtering.
It is unclear why two different models were used instead of a single model for both steps.
Additional clarification on why Alpaca was chosen over other baseline models for filtering is needed.
Dataset Size Limitation:
The benchmark contains only 500 queries, which may be insufficient for evaluating generalization across different models and settings.
A larger dataset would improve statistical robustness and generalizability.
Lack of Explicit Answer Type Differentiation:
The paper states that hallucinations are judged based on correct responses and cases where a model admits it doesn’t know.
However, there is no explicit dataset categorization distinguishing these response types.
A separate label for "acknowledging lack of knowledge" could be useful.
Small-Scale Human Evaluation in Section F:
The GPT-4 vs. human evaluation comparison in Appendix F only uses 20 samples, which is too small to draw strong conclusions.
A larger-scale human evaluation would improve confidence in the findings.
Potential Bias from GPT-4 Reference Answers:
Since GPT-4 is used to generate reference answers, its hallucination tendencies may introduce bias into the evaluation.
The paper could benefit from human-curated reference answers or a comparison between GPT-4-generated and human-generated references.]

---

### Official Review · Reviewer_ypAP · 2025-03-02
**Good explanation of the benchmark generation steps, but a more in-depth analysis of the results would be valuable**

**Rating:** 6
**Confidence:** 2

**Review:**

### Summary
- The paper proposes a benchmark for testing LLM hallucinations on "real-world" questions, mostly sourced from ShareGPT. This adds value since standard hallucination benchmarks typically focus on standardized tasks (e.g., summarization, machine translation), which do not fully represent real-world interactions with LLMs. The benchmark defines a pipeline for selecting hallucination-prone questions, categorizing them into several groups, and evaluating model hallucinations using an LLM judge.  The "ground truth" answers are generated using a RAG approach, retrieving passages from a search engine.

### Strengths
- While there may be room for improvement, the authors focus on thoroughly developing each step of the benchmark pipeline.
- The authors conduct experiments to validate the effectiveness of the RAG approach.
- The benchmark’s limitations are clearly stated.

### Weaknesses
- It is unclear to me whether the filtering approach effectively selects the most hallucination-prone questions. Some counterfactuals or examples would be valuable.
- The result analysis could be expanded. It is unclear how hallucination scores compare to non-"in the wild" questions, given different scoring scales and significant hallucination rates in both cases.
- The paper format is in two-column layout and has not been updated for ICLR.

---

### Decision · Program_Chairs · 2025-03-01

Accept